# Before Direct-Acting Antivirals for Hepatitis C Virus: Evaluation of Core Protein R70Q and L/C91M Substitutions in Chronically Infected Brazilian Patients Unresponsive to IFN and/or RBV

**DOI:** 10.3390/v15010187

**Published:** 2023-01-09

**Authors:** Letícia Bomfim Campos, Nathália Alves Araújo de Almeida, Catarina Góis de Santana, Evorah Nascimento Pereira Barbosa, Marco Aurelio Pereira Horta, Márcia Amendola Pires, Carlos Eduardo Brandão Mello, Vanessa Salete de Paula, José Júnior França de Barros

**Affiliations:** 1Laboratory of Molecular Virology, Oswaldo Cruz Institute/Fiocruz, Rio de Janeiro 21040-900, Brazil; 2Gaffrée and Guinle University Hospital, Ambulatory of Liver Disease, Rio de Janeiro State Federal University/UniRio, Rio de Janeiro 20270-004, Brazil; 3BSL-3 Facility, Oswaldo Cruz Institute/Fiocruz, Rio de Janeiro 21040-900, Brazil

**Keywords:** chronic hepatitis C, non-sustained viral response, amino acid substitutions R70Q/H and L/C91M, clinical and epidemiological profiles, host risk factors, subpopulations, pyrosequencing

## Abstract

Although chronic hepatitis C has been effectively treated with direct-acting antivirals (DAAs), the use of conventional therapy with peg-interferon (Peg-IFN) or (predominantly) ribavirin (RBV), remains widespread. R70Q/H and L/C91M amino acid substitutions in the hepatitis C virus (HCV) core protein may modulate responses to IFN and/or RBV, and are associated with cirrhosis, hepatocellular carcinoma (HCC), insulin resistance, and liver steatosis. We evaluated the R70Q/H and L/C91M substitutions, clinical and epidemiological profiles, and risk factors of Brazilian patients chronically infected with HCV subgenotypes 1a and 1b (HCV-GT1a and HCV-GT1b) unresponsive to IFN and/or RBV therapy. Sequencing and pyrosequencing analyses and sociodemographic and clinical predictive variables were used to assess the relationship between R70Q/H and L/C91M substitutions. Leukocyte counts, ALT levels, and ALT/AST ratios were significantly reduced in treated individuals, but more of these patients had advanced fibrosis and cirrhosis. L91M was more prevalent (19.7%), occurring only in HCV-GT1b, followed by R70Q/P (11.5%) and R70P (1.4%). R70Q/P exhibited higher mean AST, ALT, and GGT values, whereas L91M showed higher mean GGT values. Pyrosequencing of the L91M position revealed mutant subpopulations in 43.75% of samples.

## 1. Introduction

Approximately 58 million persons are infected with the hepatitis C virus (HCV) worldwide, and 262,815 cases of hepatitis C have been identified in Brazil [1,2]. It is estimated that 80% of those infected with HCV will develop the chronic form of the disease, of which approximately 10–15% progress to cirrhosis and 1–4% might develop hepatocellular carcinoma (HCC). HCC is a major indication for liver transplantation and the fourth leading cause of cancer death in the world [3,4]. HCV has a wide genetic variability owing to its high mutation rate, leading to the formation of viral quasispecies. It is classified into eight genotypes (GT, numbered 1–8), and several subgenotypes, defined by letters, e.g., HCV-GT1a and HCV-GT1b [5]. The global distribution of HCV genotypes and their subtypes varies according to the region. Genotype 1 is the most prevalent in the world and in Brazil (HCV-GT1a and HCV-GT1b, respectively) and is associated with more severe liver disease and a more aggressive course of infection, in addition to presenting a worse sustained virological response (SVR) to conventional treatment with peg-interferon (Peg-IFN) and ribavirin (RBV) or Peg-IFN/RBV (35–45%) [6,7].

In addition to viral replication, HCV proteins also affect various cellular functions that may be related to the pathogenesis of chronification and liver damage. The HCV core protein has a series of biological actions, such as the control of cell growth, apoptosis, oxidative stress, and immunomodulation during hepatocyte infection [8,9]. Phenotypic substitutions at position 70 (change of arginine—wild type—to glutamine—mutant type—R70Q/H) and/or 91 (change of leucine or cysteine—wild type—to methionine—mutant type—L/C91M) of the core protein are associated with insulin resistance (IR), a more severe stage of liver disease for cirrhosis, HCC, and a non-sustained virological response to Peg-IFN-RBV combination therapy [10,11,12,13,14,15,16,17].

Despite the lack of an effective vaccine, for the last few years there has been a great advance in the treatment of this infection with the development of direct-acting oral antiviral drugs (DAAs). Treatment involves shorter courses of DAAs, few adverse effects, improved tolerability, and SVR rates exceeding 95% [18]. However, restricted access to health services and the high cost of DAAs continue to prevent universal treatment replacement in low-income countries. In addition, there is still a risk of developing liver complications, such as HCC, even after successful treatment. The molecular mechanisms involved in viral and host factors have not yet been fully elucidated [19,20]. In Brazil, which comprises a heterogeneous and admixed population (multiracial population), there are no studies showing the frequency in R70Q and L/C91M. Demographic, clinical, and laboratory aspects (risk factors) of the HCV core protein in a cohort of non-responders to treatment are still to be reported. This study aimed to assess the demographic, clinical, and laboratory profiles of patients from a Brazilian (multiracial population) cohort who were unresponsive to therapy (IFN and/or RBV) associated with the R70Q/H and L/C91M substitutions, and to detect and quantify the presence of viral quasispecies.

## 2. Materials and Methods

### 2.1. Ethics 

This study was approved by the Ethics Committee of the Oswaldo Cruz Foundation (CAAE 34246914.4.1001.5248, number: 2.927.747/18). The purpose of the study was explained to participants. Confidentiality regarding patient identity and personal information was assured by highlighting that only researchers could access the information, which would be used solely for research purposes. Participants were then asked to sign an informed consent form.

### 2.2. Study Participants and Procedures

This was a cross-sectional study based on analyzed data (medical records) and laboratory test results. The samples were collected from 286 patients chronically infected with HCV (HCV-GT1a and HCV-GT1b). A single serum collection per patient was performed between November 2015 and November 2017 at the Gaffrée e Guinle University Hospital in Rio de Janeiro, Brazil.

A simple random probability sampling form was used so that participants were likely to be representative of the overall population, thus ensuring the internal validity of the study. The minimum number of subjects (*n* = 57) was determined to be N = z2 × p × (1 − p)/e2, where z was the confidence level based on a standard normal distribution (1.96 for 95%), p was the expected prevalence (0.183 for HCV in the general population, because no data were available for the study population), and e was the maximum acceptable error in the estimate (0.05) [2]. 

Inclusion criteria were an age over 18 years and confirmed hepatitis C monoinfected or coinfected with human immunodeficiency virus (HIV). The exclusion criteria were hemolyzed samples and/or a serum volume below 200 µL and the absence of clinical and laboratory information in the patients’ medical records.

### 2.3. RNA Extraction and Amplification

The number of total RNA samples was 243. RNA was extracted using a commercial High Pure Viral Nucleic Acid Kit (Roche Applied Science, Penzberg, Germany), according to the manufacturer’s instructions. The primers and thermal cycling conditions for the partial amplification of the HCV core region (starting at nucleotide minus 12 and ending at nucleotide +343, considering as position 0 the initiation codon of the open reading frame) were performed as previously established [17,21].

Briefly, RT-PCR was done with the commercial kit SuperScript^®^ III One-Step RT-PCR System with Platinum® Taq DNA Polymerase (Invitrogen, Carlsbad, CA, USA); oligonucleotides Sc2 (sense) and Ac2 (antisense) were used at 10 pmol/µL. The product of this amplification, of approximately 440 bp, was used to improve sensitivity as a template for the second amplification stage (Nested PCR) using internal and different oligonucleotides S7 (sense) and A5 (antisense) at 10 pmol/µL and the commercial kit Platinum Taq DNA polymerase (Invitrogen, Carlsbad, CA, USA). The final product had approximately 354 bp. The second round PCR products were purified (Wizard SV Gel and PCR Clean-Up System Promega, Madison, WI, USA), and quantified using a molecular mass marker (Invitrogen/Life Technologies, Carlsbad, CA, USA). 

To provide complete nucleotide sequences of the region that encodes the HCV core in genetic databases, two sets of primers were designed and used for PCR. The RT-PCR reaction was performed with the commercial kit SuperScript^®^ III One-Step RT-PCR System with Platinum^®^ Taq DNA Polymerase (Invitrogen, Carlsbad, CA, USA), and oligonucleotides HCV-F-288: 5′ ACTGCCTGATAGGGTGCTTGCG 3′ (sense) and HCV-R-1319: 5′ CCARTTCATCATCATRTCCC 3′ (antisense) were used at 10 pmol/µL. Thermocycling conditions were as follows: cDNA synthesis step, 45 °C for 30 min and 94 °C for 2 min. This was followed by five cycles of 94 °C for 1 min, 62 °C for 1 min, and 68 °C for 1 min. Another 10 cycles of 94 °C for 1 min, 60 °C for 1 min, and 68 °C for 1 min were performed. This was followed by 10 cycles at 94 °C for 1 min, 56 °C for 45 s, and 68 °C for 1 min. Another 10 cycles of 94 °C for 1 min, 50 °C for 45 s, and 68 °C for 1 min were performed. This was followed by a final extension at 68 °C for 7 min. 

The product of this amplification (1032 bp) was used as a template for the second amplification stage, using the primers HCV-F-321: 5′ AGGTCTCGTAGACCGTGCA 3′ (sense) and HCV-R-1316: 5′ RTTCATCATCATRTCCCA 3′ (antisense) at 10 pmol/µL. The RT-PCR was performed using the commercial kit Platinum Taq DNA polymerase (Invitrogen, Carlsbad, CA, USA). The cycling conditions were 94 °C for 2 min, followed by 10 cycles at 94 °C for 1 min, 62 °C for 45 s, and 72 °C for 1 min. Another 10 cycles were performed at 94 °C for 1 min, 60 °C for 45 s, and 72 °C for 1 min, followed by 10 cycles at 94 °C for 1 min, 57 °C for 45 s, and 72 °C for 1 min. A final extension was performed at 72 °C for 10 min. The final product of this PCR was approximately 996 bp. HCV-H77 (NC_004102) was used as a reference to name and position primers in the region of interest of the genome.

### 2.4. Sequencing and Molecular Analysis of the HCV Core

Purified and quantified amplicon DNA was submitted for nucleotide sequencing using the Sanger method and read from both strands using the BigDye Terminator Cycle Sequencing Ready Reaction Kit (Applied Biosystems, Foster City, CA, USA) on an ABI Prism 3730 Genetic Analyzer (Applied Biosystems, Foster City, CA, USA). The alignment, editing, and analysis of nucleotide sequences for genotyping, mutations, and deduced amino acid sequence similarities of the core gene were performed using BioEdit v7.0.5 and MEGAX programs, respectively [22,23]. The total number of amplicons obtained (HCV-GT1a and HCV-GT1b) was 208. From those, wild type, without substitution at positions 70 and 91, corresponded to 102 samples, and mutants, with R70Q and/or L91M substitution, to 106 samples. Among the 208 patients, 137 received conventional treatment and 71 were untreated.

The nucleotide sequences for the full-length protein core representing all possible substitutions analyzed in the present study have been submitted to GenBank with accession numbers, for 70H: ON563228 (HCV-GT1a_CORE_Isolate_301) and ON563229 (HCV-GT1b_CORE_Isolate_159); for 91M ON563230 (HCV-GT1b_CORE_Isolate_116); for 70Q and 91M (double mutation) ON563231 (HCV-GT1b_CORE_Isolate_120); for 70Q and 91L (double wild) ON563232 (HCV-GT1a_CORE_Isolate_49) and ON563233 (HCV-GT1b_CORE_Isolate_268). For more information and access to these sequences, use the link: https://www.ncbi.nlm.nih.gov/genbank/ (accessed on 26 November 2022).

### 2.5. Pyrosequencing 

With this methodological tool, substitution at position 91 was evaluated because it was the most prevalent substitution in the study. We included 73 HCV-GT1b samples that were treated with Peg-IFN and/or RBV and failed antiviral therapy to assess whether there would be mutant subpopulations that could influence prognosis and treatment response among the wild-type samples.

The partial sequences of the HCV core region (HCV-GT1b) obtained from GenBank were aligned to construct the respective consensus sequences. These were submitted to the PyroMarkAssay Design Software 2.0 to build specific primers (one of which was conjugated with biotin) for the amplification reactions and subsequent pyrosequencing.

For pyrosequencing, the cDNA of the RT-PCR reaction (Sc2 and Ac2) was used as a template for the second PCR, using specific primers (forward: HCV-Pyro-F-448: 5′ TGYTGCCGCGCAGGGGC 3′ and reverse: HCV-Pyro-R-636: 5′ ACAGGAGCCAYCCYGCCC 3′, the reverse-primer being conjugated to biotin). The thermal cycling conditions were an initial denaturation at 94 °C for 2 min, followed by 35 cycles of 94 °C for 1 min, 62 °C for 45 s, and 72 °C for 1 min, and a final extension at 72 °C for 10 min. The generated 188 bp fragment was used as a template for pyrosequencing. The order of nucleotide dispensing was (A/C/T) T (T/C/A/G) for the L91M variation, and the forward primer HCV-F-PyroSeq-91aa: 3′ CTCTATGGCAAYGAGGGY 5′ was used for the pyrosequencing reactions. The following steps were performed using PyroMark Q96 ID equipment (QIAGEN, Hilden, Germany) [24].

### 2.6. Statistical Analysis

The data are presented as frequencies and percentages for categorical variables and as means with standard deviations for continuous variables. The X^2^ test, Mann–Whitney U test, Student’s *t*-test, or Wilcoxon’s test was used to analyze categorical, parametric continuous, and non-parametric variables, as appropriate. Associations between treated and untreated individuals, as well as mutant and non-mutant were performed against several potential predictor variables (sex, body mass index [BMI], fasting blood glucose, triglyceride, total cholesterol, low-density lipoprotein [LDL], high-density lipoprotein [HDL], aspartate aminotransferase [AST], alanine aminotransferase [ALT], gamma glutamyl transferase [GGT] levels, fibrosis degree, steatosis, and platelet level).

## 3. Results

### 3.1. Characteristics of the Study Population

In 286 patients, we reported on 127 male and 159 female patients, with a mean age of 60.7 ± 10.2 (range: 32–86) years and a progressive increase in the number of individuals with increasing age group. In our study, 40.2% (115/286) were white, 24.1% (69/286) were black, and 21.0% (60/286) were mixed race; 4.7% (42/286) omitted that answer from the questionnaire during anamnesis. Most individuals had fibrosis stage F4/cirrhotic (59.4% or 170/286), and the presence of HCC was low, at 2.1% (6/286). Most patients were overweight (31.5%, 90/286) or obese (16.1%, 46/286). In 12.5% (36/286) of the patients, we detected HCV/HIV coinfections. Regarding HCV genotyping by phylogenetic analysis (data not shown), 52.8% (151/286) were HCV-GT1a and 47.2% (135/286) were HCV-GT1b.

From the 286 patients included in the study, 171 patients failed conventional therapy based on Peg-IFN and/or RBV and 115 patients did not undergo antiviral treatment. The demographic and biochemical data of the treated and untreated study population are summarized in Table 1. The dose of Peg-IFN therapy in the treated group was Peg-IFN alpha 2a containing 180 micrograms, injectable solution, and Peg-IFN alpha 2b at 0.5 to 1.0 micrograms per kilogram of patient weight, injectable solution. These solutions were administered once a week, subcutaneously, for 48 weeks, in monotherapy or associated with RBV. The dose of RBV (250 mg) varied according to the patient’s body weight. On average, 750 to 1250 mg/day were administered orally.

### 3.2. Frequency of Substitutions R70Q and L91M

In total, 208 samples were sequenced, of which 137 were treated and 71 were untreated. Table 2 shows the frequency of substitutions and the demographic, biochemical, and histological characteristics of patients with chronic hepatitis C. There were no variables that were significantly different between the groups. 

The most frequent substitution was at position 91 (L91M), found in 19.7% (41/208) of samples and occurring only in HCV-GT1b. The R70Q/P variation was found in 11.5% (24/208) of the study population, and the frequency of both substitutions (R70Q/P and L91M) occurring concomitantly was 19.7% (41/208). We found that R70P (the substitution of an arginine for a proline at amino acid position 70) had a frequency of 1.4% (3/208).

From all the sequenced samples, 106 HCV-GT1a and 102 HCV-GT1b samples were included in these analyses. Patients infected with HCV-GT1a had a lower frequency of substitutions, with only R70Q/P found in 13.2% (14/106). HCV-GT1b showed higher frequencies of all the substitutions investigated, with L91M being the most frequent, found in 40.2% (41/102), followed by R70Q/P in 9.8% (10/102). The prevalence of a double variation (L91M and R70Q occurring simultaneously) was 19.7% (41/208) and was found only in HCV-GT1b (Table 2).

### 3.3. Pyrosequencing Assays

Pyrosequencing of the substitution L91M position in 73 samples of HCV-GT1b treated with IFN and/or RBV was performed to identify subpopulations. In this analysis, we included two groups: the first with 16 samples classified as wild type by Sanger sequencing and the second with 57 samples classified as mutants by Sanger sequencing. 

Figure 1 shows a comparison of the results obtained by Sanger sequencing (Figure 1A) and pyrosequencing (Figure 1B). The overlap of nucleotides was evident, where the traditional sequencing identified the most prevalent nucleotide at that position, a thymine (T), which encodes for leucine (wild-type). However, a subpopulation with an adenine (A), which encodes methionine (mutant) was observed. In contrast, pyrosequencing enabled identification and quantification of the nucleotide populations, facilitating better analysis and monitoring of mutations. This difference in the results provided by the sequencing methods (Figure 1) was responsible for standardization of the pyrosequencing technique for the detection of L91M in the HCV core. 

In general, HCV subpopulations were detected in 37.0% (27/73) of the analyzed samples by pyrosequencing. Among the 57 samples classified as mutants by Sanger sequencing, wild-type subpopulations were found in 35.1% (20/57) by pyrosequencing. In the group comprising 16 samples classified as wild type by Sanger sequencing, mutant subpopulations were found in 43.75% (7/16) of the samples by pyrosequencing (Table 3). All pyrosequencing results are available in the Appendix A.

## 4. Discussion

Hepatitis C is a serious public health challenge associated with the development of cirrhosis and HCC [1]. Several studies revealed an association between amino acid substitutions at the R70Q/H and L91M positions with disease progression to cirrhosis and HCC and a poor response to treatment with Peg-IFN and/or RBV [10,11,12,13,14,15,16,17]. 

However, DAAs have revolutionized the modern treatment of chronic hepatitis C. The current therapeutic protocol for HCV with DAAs is not widely accessible due to the high cost of DAAs, particularly in underdeveloped countries; therefore, many countries use IFN-based therapies, with or without RBV, and in specific clinical conditions, RBV is an optional component [18]. 

In our study, the L91M substitution was more frequent (19.7%, 41/208), and it was found only in HCV-GT1b. The presence of the cysteine residue at position 91 in the core protein of HCV-GT1a suggests a molecular signature unique to this subgenotype, as only one study reported the presence of the C91M variation in HCV-GT1a [27]. The substitution of the amino acid cysteine to methionine requires a nucleotide substitution in the three nitrogenous base positions (triplets) that form the genetic codon for this amino acid, which seems to us harder to happen from the point of mutation rate. In our analysis, the mean level of the liver enzyme GGT was higher in individuals with the L91M mutation than in wild-type individuals, but statistical differences were probably due to the low number of individuals. 

The R70Q/P variation was found in 11.5% (24/208) of patients; these substitutions are associated with a worse clinical stage and a greater progression of the disease to cirrhosis and HCC [10,11,12,13,14,15,16,17]. In our analysis, R70Q/P showed higher levels of mean AST, ALT, and GGT enzymes, which are important biochemical markers for the progression of HCV and SVR [28]; however, no statistical differences were found, possibly due to the sample size not being larger. The frequency of both substitutions (R70Q/P and L91M) was 19.7% (41/208), and the mean glucose level in the mutant group was higher compared to wild-type individuals, although no statistical differences were found. HCV modulates hepatic glucose metabolism by impairing insulin signaling and glucose uptake [29]. This substitution is related to the presence of insulin resistance (IR) and glucose intolerance, mainly in Asian individuals, although a lack of association with IR has been reported in Brazilian patients [12,30,31,32]. 

In this study, we identified three individuals with R70P, two HCV-GT1a and one HCV-GT1b; one R70P individual presented with cirrhosis (F4); and all three had steatosis. This substitution has only been previously reported in Argentinian and Swedish patients, but no association has been reported [28,33]. A meta-analysis showed that the frequency of 70P was 3% in genotype 3 and 13% in genotype 6 [28]. Therefore, the frequency of this variation was extremely low, especially for genotype 1. This study reports R70P in Brazilian patients for the first time.

HCV-GT1b showed higher frequencies for the two substitutions studied, with L91M being found in 40.2% (41/102) and R70Q/P in 9.8% (10/102) of individuals. These data suggest that HCV-GT1b has a higher frequency of these mutations than that in HCV-GT1a [15]. In addition to having a higher frequency of both substitutions, it is known that HCV-GT1b is associated with the rapid progression of cirrhosis and HCC [34,35,36]. As the number of individuals with HCC was small in our study, we did not find a statistically significant relationship between the studied mutations and the presence of liver cancer (HCC). In Brazil, a study published by our team demonstrated that a substitution at position R70Q was significantly more frequent in patients with cirrhosis and HCC than in non-cirrhotic and non-HCC individuals [17].

The collection of samples from patients who did not respond to conventional treatment, RBV or Peg-INF/RBV, and from patients naïve to treatment were performed upon invitation to participate in the treatment with DAAs. As such, in this cross-sectional study it is not possible to state that the adopted treatment acted as a selective pressure event allowing the rapid emergence of new viral variants presenting adaptive advantages inherent to their evolutionary biology. Previous studies indicate that treatment with IFN and/or RBV can induce selective pressure allowing treatment-resistant HCV variants to become major subpopulations. However, to check for such evolution, this study would have needed to follow these individuals before, during and after treatment with IFN and/or RBV. 

Most patients included in the study were obese or overweight, and these conditions tend to be an aggravating factor for a worse prognosis of chronic HCV infection. Although we did not observe significant values for IR and steatosis in our study population, it is known that R70Q is associated with these conditions. The R70Q substitution is related to the increased expression of IL-6, which may cause IR, steatosis, and HCC and inhibit IFN signaling, which is associated with therapeutic failure [37].

In the analysis of nucleotide substitutions, we observed the presence of mutant subpopulations in 43.75% of the samples classified as "wild type samples" according to the Sanger sequencing results. Pyrosequencing to monitor mutations associated with clinical inflammatory evolution and factors related to SVR confirmed and quantified quasispecies fluctuations. These minority (mutant) subpopulations could become the majority and are implicated in therapy and natural history, leading to further progression to cirrhosis and HCC [38].

The statistically significant decrease of some markers of the clinical evolution of HCV infection, such as AST, AST/ALT ratio, and defense cells in patients treated with the conventional protocol, may suggest a slight or transient improvement, even if SVR is not achieved. This study showed a significant reduction in the number of leukocytes in individuals who received conventional treatment when compared to naive patients. The reduction in the number of leukocytes was likely due to conventional treatment, as the number of coinfected patients (HCV/HIV) receiving antiretroviral therapy was relatively small and was present in both groups.

Patients with advanced stages of liver damage caused by the infectious inflammatory HCV processes may continue to present an important risk for progression of the fibrotic condition and development of HCC, even after clearance of the viral infection. SVR can reduce the risk of developing HCC in most patients with HCV-related liver cirrhosis; however, some studies have reported an increased and recurrent rate of HCC in patients with HCV cirrhosis who had been treated with DAAs [39,40,41]. The mechanisms underlying these events are not fully understood [42]. 

Our results might be influenced by the fact that the included patients are older and mainly men, belonging to the so-called baby boomer generation, given that the Brazilian epidemiological data associates these characteristics with the absence of HCV detection tests. From 1992 onwards, a screening process was implemented in Brazil that allowed the detection of HCV in blood samples.

## 5. Conclusions

This is the first demographic and clinical laboratory profile study associated with R70Q/P and L/C91M substitutions in a group of Brazilian patients chronically infected with HCV-GT1a and HCV-GT1b who failed conventional therapy. This highlights the importance of these substitutions (R70Q/P and L91M) in understanding the natural history of HCV infection. Although we did not find statistical differences due to the sample size that was a limiting factor in the group with mutations, we observed higher averages of important biochemical markers of hepatic progression. In 2015, Brazil implemented the Clinical Protocol and Therapeutic Guidelines for Hepatitis C and Coinfections, starting with the use of DAAs [43] After the sample collection period, the patients included in the study received DAAs, and 99.3% achieved SVR. This shows that broad access to DAAs therapy is an important way to eradicate HCV and must be collectively and globally addressed as a public health policy and strategy to be implemented to combat and effectively treat hepatitis C infection, as well as its subsequent clinical complications.

This study highlights not only the clinical-laboratory and virological profile of patients who do not respond to IFN and/or RBV therapy, who were submitted to a new pharmacological treatment, but also reports the efficiency of the new DAAs as a policy of public health targeting HCV eradication. This study is relevant in Brazil and may be useful in the future for a follow-up study of patients eligible for the new treatment.

The development and use of methodologies capable of revealing the natural emergence of therapy-resistant mutants, early detection of viral genetic polymorphisms, and constant monitoring of patients after treatment constitute critical prognostic tools to reduce the likelihood of hepatic inflammatory progression and could improve the clinical conditions of the patient.

## Figures and Tables

**Figure 1 viruses-15-00187-f001:**
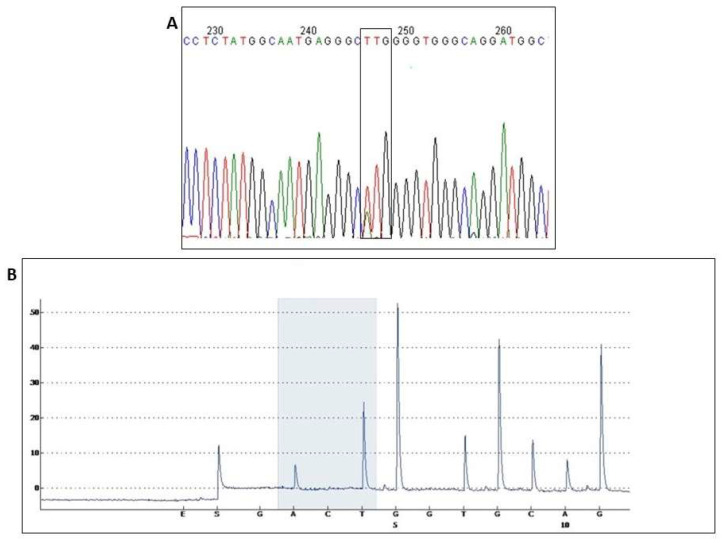
Results obtained by Sanger sequencing and pyrosequencing of sample 94. Chromatograms and pyrogram ((**A**,**B**) respectively), with obvious nucleotide fluctuations (**A**), a thymine (T), which translates to leucine and an adenine (A) which encodes methionine overlap. Pyrosequencing confirms and quantifies this variation, which shows a slight difference between the presence of the two subpopulations (T nucleotide, wild type; and A, mutant type).

**Table 1 viruses-15-00187-t001:** Demographic and biochemical characteristics of the study population with and without treatment (mean ± SD and case number).

Variable	Treated (*n* = 171)	Untreated (*n* = 115)	*p*-Value
Age, years	60.6 ± 9.0	61.0 ± 11.7	0.13
Female sex	90 (52.6%)	69 (60%)	0.11
Body weight, kg	71.9 ± 13.1	69.7 ± 13.7	0.19
BMI, kg/m^2^	26.4 ± 4.3	26.0 ± 4.9	0.24
Albumin, mg/dL	4.04 ± 0.63	4.04 ± 0.76	0.09
Fasting blood glucose, mg/dL	107.2 ± 31.6	106.6 ± 35.1	0.80
Triglyceride, mg/dL	111.8 ± 60.1	106.8 ± 48.3	0.36
Total cholesterol, mg/dL	159.6 ± 32.9	159.9 ± 33.4	0.98
LDL, mg/dL	87.2 ± 28.7	89.7 ± 56.45	0.51
HDL, mg/dL	51.3 ± 28.7	63.4 ± 65.9	0.34
AST, U/L	**60.7 ± 43.4**	**67.9 ± 56.6**	**0.03**
AST/ALT ratio	**1.1 ± 0.5**	**1.8 ± 0.6**	**0.02**
Platelet count/L	128 × 10^9^ ± 110 × 10^9^	139 × 10^9^ ± 92 × 10^9^	0.38
GGT, U/L	100.2 ± 96.9	103.03 ± 106.2	0.58
Total bilirubin, mg/dL	0.86 ± 0.62	0.88 ± 0.58	0.24
Unconjugated bilirubin, mg/dL	0.51 ± 0.40	0.52 ± 0.31	0.78
Hemoglobin, g/L	13.41 ± 2.60	13.08± 2.09	0.32
Leukocytes, cells/mm³	**4.2 × 10^3^ ± 2.8 × 10^3^**	**5.3 × 10^3^ ± 3.6 × 10^3^**	**0.01**
Hematocrit	40.0% ± 5.0%	39.0% ± 5.0%	0.39
ALT, U/L	62.43 ± 58.46	65.0 ± 47.15	0.74
AFP, ng/mL	16.28 ± 24.12	14.32 ± 23.18	0.63
Elastography, kPa	18.43 ± 12.34	18.15 ± 12.57	0.89
Steatosis	75/130 (57.6%)	55/130 (42.3%)	0.96
HCC	3/6 (50%)	3/6 (50%)	0.79
Fibrosis Degree			**0.004**
F1	10/15 (66.6%)	5/15 (33.4%)	
F2	19/23 (82.6%)	4/23 (19.4%)	
F3	42/78 (53.8%)	36/78 (46.2%)	
F4	101/170 (59.4%)	69/170 (40.6%)	
Subgenotype			0.92
1a	95/151 (62.9%)	56/151 (37.1%)	
1b	76/135 (56.2%)	59/135 (43.8%)	
Only amino acid 70 mutant	16/24 (66.7%)	8/24 (33.3%)	0.12
Only amino acid 91 mutant	26/41 (63.4%)	15/41 (36.6%)	0.13
Amino acid 70 and 91 mutant	29/41 (70.7%)	12/41 (29.3%)	1.0

Bold values are significant at *p* < 0.05 or have marginal significance (*p* < 0.10). Reference values: Previous treatment performed with interferon and/or ribavirin; F1: mild fibrosis; F2: moderate fibrosis; F3: advanced fibrosis; F4: cirrhosis; BMI (body mass index, kg/m^2^); Normal: 18.5–25, overweight: 25–30, obese > 30; AST (U/L) 5–40: normal, >40: high; ALT (U/L) 7–56: normal, >56: high; GGT (U/L) male 8–61: normal, >61: elevated, female 5–36: normal, >36 elevated; BT (mg/dL) up to 1.2 L: normal; albumin (g/dL) 3.5–4.7: normal, <3.5: low, >4.7 high; total cholesterol (mg/dL), up to 190 normal; >190, high [25,26]. SD, standard deviation; LDL, low-density lipoprotein; HDL, high-density lipoprotein; AST, aspartate aminotransferase; ALT, alanine aminotransferase; GGT, gamma-glutamyltransferase; AFP, alpha fetoprotein; HCC, hepatocellular carcinoma.

**Table 2 viruses-15-00187-t002:** Demographic, biochemical, and histological characteristics of chronic hepatitis C patients with different substitutions in the core region (mean ± SD and case number).

Variable	aa70 (*n* = 24)	aa91 (*n* = 41)	aa70 and 91 (*n* = 41)	Wild (*n* = 102)
Age, Years	64.09 ± 8.5	62.05 ± 8.5	64.9 ± 6.9	58.8 ± 10.3
Male sex, n (%)	9/24 (37.5)	21/41 (51.2)	11/41 (26.8)	56/102 (54.9)
BMI, kg/m^2^	27.02 ± 4.8	26.0 ± 4.7	25.6 ± 3.6	26.1 ± 4.8
Albumin, mg/dL	3.98 ± 0.8	4.26 ± 0.9	3.96 ± 0.7	4.87 ± 5.7
Fasting blood glucose, mg/dL	123.9 ± 44.3	95.14 ± 19.1	111.16 ± 33.6	104.14 ± 25.43
Triglyceride, mg/dL	111.0 ± 48.0	102.46 ± 43.9	108.75 ± 88.1	108.69 ± 47.7
Total cholesterol, mg/dL	163.27 ± 27.1	168.5 ± 34.8	158.85 ± 33.1	159.15 ± 33.8
LDL, mg/dL	93.17 ± 29.8	94.11 ± 28.9	79.26 ± 33.3	87.33 ± 30.1
HDL, mg/dL	46.55 ± 11.9	55.09 ± 16.7	51.71 ± 12.6	54.10 ± 32.3
AST, U/L	83.9 ± 95.2	55.95 ± 45.3	54.9 ± 34.5	60.83 ± 40.4
Platelet count, /L	139 × 109 ± 80×109	187 × 109 ± 129 × 109	149 × 109 ± 69 × 109	155 × 109 ± 780 × 109
GGT, U/L	133.74 ± 167.6	164.3 ± 521.7	99.1 ± 104.6	109.82 ± 132.5
Total bilirubin, mg/dL	0.92 ± 0.47	0.74 ± 0.49	0.74 ± 0.44	0.92 ± 0.64
Conjugated bilirubin, mg/dL	0.37 ± 0.23	0.31 ± 0.25	0.38 ± 0.28	0.39 ± 0.32
Unconjugated bilirubin, mg/dL	0.48 ± 0.25	0.53 ± 0.39	0.51 ± 0.45	0.50 ± 0.37
Hemoglobin, g/L	12.9 ± 1.8	13.3 ± 2.0	12.6 ± 2.2	13.7 ± 2.6
Leukocytes, cells/mm³	5.0 × 103 ± 1.9 × 103	5.7 × 103 ± 2.0 × 103	5.2 × 103 ± 1.6 × 103	5.8 × 103 ± 2.9 × 103
Hematocrit, %	40.0 ± 6.0	41.0 ± 5.0	40.0 ± 4.0	40.0 ± 5.0
ALT, U/L	88.7 ± 95.7	58.7 ± 54.2	47.1 ± 32.3	60.6 ± 42.1
AFP, ng/mL	21.7 ± 23.6	6.5 ± 7.6	20.7 ± 39.5	17.4 ± 27.7
Elastography, kPa	17.4 ± 7.9	16.3 ± 9.8	20.6 ± 16.3	18.4 ± 11.5
Steatosis, n (%)	13/24 (54.2)	17/41 (41.5)	19/41 (46.3)	46/102 (45.1)
HCC	0/24 (0.0)	0/41 (0.0)	1/41 (2.4)	5/102 (4.9)
Fibrosis Degree, n (%)				
F1	0/24 (0.0)	4/41 (9.8)	1/41 (2.4)	8/102 (7.8)
F2	2/24 (8.3)	6/41 (14.6)	5/41 (12.2)	7/102 (6.9)
F3	8/24 (33.3)	10/41 (24.4)	9/41 (21.9)	27/102 (26.5)
F4	14/24 (58.3)	21/41 (51.2)	26/41 (63.4)	60/102 (58.8)
Subgenotype, n (%)				
1a	14/24 (58.3)	0/41 (0.0)	0/41 (0.0)	92/102 (90.2)
1b	10/24 (41.7)	41/41 (100.0)	41/41 (100.0)	10/102 (9.8)
Treatment, n (%)				
Yes	16/24 (66.7)	26/41 (63.4)	29/41 (70.7)	66/102 (64.7)
No	8/24 (33.3)	15/41 (36.6)	12/41 (29.3)	36/102 (35.3)

Reference values: Previous treatment performed with interferon and/or ribavirin; F1: mild fibrosis; F2: moderate fibrosis; F3: advanced fibrosis; F4: cirrhosis; BMI (body mass index); Normal: 18.5–25, overweight: 25–30, obesity >30; AST (U/L)—5–40: normal, >40: high; ALT (U/L) 7–56: normal, >56: high; GGT (U/L) male 8–61: normal, >61: elevated, female 5–36: normal, >36 elevated; BT (mg/dL) up to 1.2 L: normal; albumin (g/dL) 3.5–4.7: normal, <3.5: low, >4.7: high; total cholesterol (mg/dL), up to 190 normal; >190, high [25,26].

**Table 3 viruses-15-00187-t003:** Pyrosequencing result in mutants and wild-types.

Subpopulations by Pyrosequencing	Rating by Sanger Sequencing	
	Wild Type Group (91L) *n* = 16	Mutant Group (91M) *n* = 57
Wild population (91L)	11/16 (68.75%)	20/57 (35.1%)
Mutant population (91M)	7/16 (43.75%)	37/57 (64.9%)

## Data Availability

The data presented in this study are openly available in GenBank. This data can be found here: https://www.ncbi.nlm.nih.gov/genbank/ (Accessed on 26 November 2022). Accession numbers: ON563228 (HCV-GT1a_CORE_Isolate_301) and ON563229 (HCV-GT1b_CORE_Isolate_159) for R70H, ON563230 (HCV-GT1b_CORE_Isolate_116) for L91M, ON563231 (HCV-GT1b_CORE_Isolate_120) for both substitutions, and ON563232 (HCV-GT1a_CORE_Isolate_49) and ON563233 (HCV-GT1b_CORE_Isolate_268).

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
