# Peer review of "Before Direct-Acting Antivirals for Hepatitis C Virus: Evaluation of Core Protein R70Q and L/C91M Substitutions in Chronically Infected Brazilian Patients Unresponsive to IFN and/or RBV"

_viruses, 2023, doi:10.3390/v15010187_

Round 1

Reviewer 1 Report (Previous Reviewer 2)

I see that the authors have done a good job improving their manuscript, and I think that the current version can be published after some proofreading.

Author Response

Comments and Suggestions for Authors

I see that the authors have done a good job improving their manuscript, and I think that the current version can be published after some proofreading.

We would like to thank you for the careful review of our article. The review of the writing in English was conducted by an internationally credible company, as per the attached certificate.

Reviewer 2 Report (New Reviewer)

The paper is well writen but it deals with an old treatment for hepatitis C that is not anymore indicated. It has an historic value, as a lot of work was carried out by the researchers. The two cited mutations have been associated with HCC but this was  not confirmed as such in other studies involving HCV and HCC including in our country.

Author Response

Comments and Suggestions for Authors

The paper is well writen but it deals with an old treatment for hepatitis C that is not anymore indicated. It has an historic value, as a lot of work was carried out by the researchers. The two cited mutations have been associated with HCC but this was not confirmed as such in other studies involving HCV and HCC including in our country.

Thank you for reviewing our article and for your comments. As the number of individuals with HCC was small in our study, we did not find a statistically significant relationship between the studied mutations and the presence of liver cancer. The review of the writing in English was conducted by an internationally credible company, as per the attached certificate. This information has now been added to the article.

This manuscript is a resubmission of an earlier submission. The following is a list of the peer review reports and author responses from that submission.

Round 1

Reviewer 1 Report

Campos et al. presented an evaluation of two key substitutions in HCV core protein, R70Q and L91M, in Brazilian patients upon interferon and ribavirin co-treatment but not even treated with direct-acting antivirals. The novelty of this study is made of two parts: patient cohort in Brazil and L91M is merely present in HCV-genotype1b but not 1a, as both mutations have been well characterized in earlier studies.

Major comments:

(1) Interferon unresponsiveness relates to IL-28B genetic polymorphism, which has played a role in the development of two mutations in the HCV core protein shown in this study. Have the authors compared IL-28B gene in the patient naive treated or treated with interferon and/or ribavirin, and implemented this as a key player in the tables?

(2) Whether interferon and/or ribavirin treatment increase the mutation rate? For instance, whether the proportion of mutated genome has increased during treatment? This can be determined by pyrosequencing samples from the same patient collected over time. 

(3) Why L91M substitution only occurs in HCV genotype 1b in Brazilian patients?

Minor comments:

(1) Interferon (IFN) is often miswritten as INF in lines 47, 57, 63, 182, 185, 241, 271, 275, 314.

(2) Why has the number of patients decreased from 286 (line 84) to 243 (line 99) to 208 (line 143)? Assuming that some were excluded due to a low amount of sample (lines 95-96) and the latter the authors were not able to sequence some patient samples, correct?

(3) The authors used nested PCR in section 2.3, which could be clearly written.

(4) a typo of QIAGEN in line 168.

(5) What is the dose and duration of interferon and/or ribavirin therapy in the cohort?

(6) Was the reduction of leukocyte number in treated patients compared to untreated shown in Table 1 due to directly antiviral therapy or HIV coinfection?

(7)  How about the mutational polymorphism at the nucleotide level in the patients, referring to lines 224-230.

(8) Whether treatment induces mutation rate in Table 2.

Reviewer 2 Report

The problem of HCV resistance against treatment was extremely acute in the 1990s and early 2000s, but after the effective direct antiviral agents (DAA) based on the inhibitors of HCV enzymes (NS3/4A protease, NS5A polymerase cofactor and NS5B polymerase) were developed, the treatment of HCV using those drugs is successful in more that 99% of cases. Unfortunately, the high price of the DAA often makes them inaccessible for patients in developing countries, and that is the reason why the studies focused on molecular monitoring of the resistant HCV genome variants are still actual.

Possible association of  R70Q and L/C91M with HCV resistance against treatment using pegylated interferon and ribavirin combination was described by Akuta N et al (Intervirology. 2005;48:372–380) and during the following years confirmed by various researchers, although the mechanism of such resistance remains unknown. The findings described in manuscript mostly correspond with literature, as authors avow  in the Discussion section.

The very important information Authors placed in the Conclusions section - After the sample collection period, 346 patients included in the study received DAAs, and 99.3% achieved SVR. I think it must be emphasized as  the message to the public health functionaries, pointing out that providing the treatment  with effective  Direct-Acting Antivirals to all HCV patients is a way for eradication of the HCV infection.

Reviewer 3 Report

This manuscript has mentioned about the evaluation of core protein R70Q and L/C91M substitution of hepatitis C virus. This is very interesting when this manuscript is published in 10 years before. It was important when HCV was treated with peg-IFN, but nowadays DAAs are major treatment with HCV infection. I think this report has no novelty and ,regret to say, is meaningless.